# Pi3DGS: Robust Joint Optimization of Camera Poses and 3DGS from Uncalibrated Images

## Abstract

Recent advancements in neural rendering for 3D reconstruction have focused on constructing representations directly from uncalibrated RGB images, bypassing the need for Structure-from-Motion (SfM) preprocessing. A primary challenge in this domain is the joint optimization of scene geometry and camera parameters, a task fraught with inherent ambiguities. Although 3D Gaussian Splatting (3DGS) has achieved photorealistic reconstruction quality, its discrete, point-based representation complicates this joint optimization process. To address these challenges, we propose a robust, SfM-free framework that leverages pre-trained 3D feed-forward models within a coarse-to-fine alignment pipeline. Our method introduces Pi3 for scene initialization and proceeds with the joint training of geometry and camera poses. To enhance the stability of camera pose optimization, we employ 3D and 2D filters to regularize the gradients from signal alignment. Furthermore, we incorporate a geometric regularization based on image matching to provide global constraints for camera pose refinement, which significantly improves both reconstruction quality and pose estimation accuracy. Our method achieves competitive performance in novel view synthesis and camera pose estimation, demonstrating its robustness across diverse datasets.

## 1 Introduction

Neural Radiance Fields (NeRF) (Mildenhall et al., 2021) provide a practical framework for learning a neural 3D radiance representation from collections of 2D RGB images, and have motivated a variety of alternative scene representations. NeRF accepts 3D sample points and view directions as input and predicts radiance and density via a neural network, which are then integrated by volume rendering. This implicit representation is a principal contributor to the high computational cost of NeRF-like methods. Recently, 3D Gaussian Splatting (3DGS) (Kerbl et al., 2023) has emerged as an explicit alternative: it represents a scene as a set of 3D Gaussian primitives and leverages conventional rasterization pipelines on modern GPUs, yielding substantially faster training and rendering.

Most reconstruction methods, including NeRF and 3DGS, require accurate camera intrinsics and extrinsics for the input images. These parameters are commonly obtained through Structure from Motion (SfM) tools such as COLMAP (Schonberger & Frahm, 2016). However, classical SfM pipelines have limitations: they depend on sufficient image overlap to establish multiview correspondences and can fail in challenging scenarios. Recent large multitask models for 3D vision, such as MASt3r (Duisterhof et al., 2025) and VGGT (Wang et al., 2025a), demonstrate improved precision and robustness across diverse conditions. Motivated by these advances, we adopt Pi3 (Wang et al., 2025c), a fully permutation-equivariant architecture, to replace SfM as a preprocessing module.

Directly using the outputs of learned large models introduces additional challenges. Network-based approaches can infer 2D–3D relations at a high level, but predicting precise numerical camera parameters (e.g., view matrices) remains difficult in the absence of strict geometric constraints (Wang et al., 2023a). Inaccurate camera parameters often lead to geometric and photometric degradations—such as misalignment, skew, or distortion—particularly in scenes with large depth ranges (e.g., outdoor environments or long image sequences). To address this, we propose Pi3DGS, a joint training framework that reconstructs 3DGS geometry while simultaneously refining camera poses, starting from the imperfect camera estimates produced by Pi3.

Adapting joint optimization strategies from NeRF to 3DGS entails specific difficulties. First, 3DGS employs an explicit representation composed of Gaussian ellipsoids whose attributes are optimized and whose population is incrementally densified to recover higher-frequency detail. This densification process splits or duplicates primitives when they exceed a threshold. Because 3DGS is prone to overfitting, primitives created under supervision of inaccurate early-stage cameras can become trapped in local minima, and there is no inherent mechanism to correct such primitives later in training.

To mitigate early densification artifacts, and inspired by Mip-Splatting (Yu et al., 2024), we introduce a gradient-smoothing strategy. The key idea is to temporarily scale or blur the Gaussians so that each primitive influences more pixels during early training, thereby smoothing gradients and emulating the coarse-to-fine behavior observed in AbsGS (Ye et al., 2024). Consistent with analyses in TensorRF (Chen et al., 2024), this attenuation of high-frequency signals yields more stable camera alignment and reduces the tendency to converge to poor local minima.

A second challenge is the relative paucity of cross-view constraints during camera optimization. NeRF-based methods typically sample rays from multiple images per batch, enabling simultaneous optimization across many cameras and enforcing multi-view consistency. In contrast, 3DGS uses a differentiable rasterization pipeline in which all primitives are transformed, clipped, rasterized, and blended to produce pixel colors; this pipeline limits the number of views that can be jointly optimized per step and, absent explicit geometric coupling, makes cameras susceptible to inconsistent local groupings.

To provide stronger geometric supervision, and drawing on PoRF (Bian et al.), we integrate an image-matching-based geometric constraint into the pose refinement process. This constraint supplements the rasterization loss, stabilizes camera optimization (in particular during opacity resetting), and improves final pose accuracy.

Our main contributions can be summarized as follows.

- We present Pi3DGS, a pipeline for reconstructing 3D Gaussian Splatting scenes from RGB images alone, obviating the need for accurate SfM precomputation. The pipeline combines a learned permutation-equivariant initialization (Pi3) with a joint optimization that constructs the 3DGS representation while simultaneously refining camera intrinsics and extrinsics.

- We introduce a filtering strategy based on gradient smoothing to prevent premature and erroneous densification of Gaussian primitives. By enlarging the effective receptive field of primitives during early training and attenuating high-frequency signals, this strategy mitigates unstable gradients and reduces convergence to poor local minima, thereby enhancing training stability.

- We incorporate image-matching geometric constraints into the pose-refinement stage to enforce robust multi-view consistency. This regularization complements the rasterization loss, stabilizes camera optimization (particularly during opacity resetting), and improves final pose accuracy and scene fidelity.

## 2 RELATED WORKS

### 2.1 NOVEL VIEW SYNTHESIS

Novel view synthesis is a subfield of 3D reconstruction that focuses on producing high-quality images from viewpoints not present in the input image set. This topic is closely related to inverse differentiable rendering and other recent trends in neural rendering. Neural Radiance Fields (NeRF) (Mildenhall et al., 2021) marked a milestone by demonstrating the effectiveness of implicit neural fields for photorealistic novel view synthesis. Subsequent approaches based on neural signed distance functions (SDFs) replace radiance-based volumetric representations with surface-centric formulations (Wang et al., 2021; 2023b; Li et al., 2023), which can be more readily converted to classical 3D representations such as triangle meshes. Other works organize scene representations on structured voxel grids to improve efficiency (Deng & Tartaglione, 2023; Takikawa et al., 2021; Reiser et al., 2021).

Although MLP-based methods have achieved impressive visual fidelity, they face inherent challenges in representing high-frequency signals (e.g., complex textures and fine geometric detail) and often require substantial training time. 3D Gaussian Splatting (3DGS) (Kerbl et al., 2023) addresses some of these limitations by using an explicit representation that models geometry with 3D Gaussian primitives and encodes appearance via spherical harmonics (Atkinson & Han, 2012). By leveraging conventional rasterization pipelines, 3DGS attains excellent visual quality while enabling real-time rendering performance. Consequently, 3DGS has attracted considerable attention and has been applied to diverse tasks, including autonomous driving (Zhou et al., 2024), deformable human modeling (Moreau et al., 2024), and embodied intelligence training (Wang et al., 2025b).

Despite these advances, most novel view synthesis methods still depend on accurate camera poses to construct reliable scene representations; obtaining such poses remains a critical practical bottleneck in many real-world scenarios.

## 2.2 Joint Refinement on Camera and Scene

The feasibility of jointly optimizing scene representation and camera poses was first demonstrated by BARF (Lin et al., 2021), which trains a neural radiance field from RGB images given only inaccurate initial camera poses. TensoRF (Chen et al., 2024) introduces Gaussian kernel filtering of 2D supervision to smooth the optimization landscape for neural radiance fields, thus mitigating local minima in joint camera–scene optimization. Subsequent work has incorporated additional priors and geometric cues to stabilize pose estimation: Nope-NeRF (Bian et al., 2023) uses depth maps as anchors; SC-NeRF (Jeong et al., 2021) and PoRF (Bian et al.) integrate image matching to refine camera poses.

Recent methods based on 3D Gaussian Splatting (3DGS) aim to address these limitations by leveraging rasterization-friendly primitives. CF-3DGS (Fu et al., 2024) incrementally adds Gaussians and cameras from a sequence of consecutive frames, but fixes registered cameras in later stages, which can lead to accumulated registration errors; moreover, when its COLMAP-free template heuristics fail, CF-3DGS requires precomputed camera intrinsics. InstantSplat (Fan et al., 2024) employs DUSt3r (Wang et al., 2024) for initialization but does not scale well to long image sequences. 3R-GS (Huang et al., 2025) proposes an alternative pipeline (see the original reference for details). BAD-Gaussians (Zhao et al., 2024) specifically target scenes affected by motion blur by assigning and optimizing multiple virtual cameras per view to model blur trajectories. KeyGS (Chang et al., 2025) adapts the smoothing strategy from TensorRF to 3DGS through Mip-Splatting (Yu et al., 2024), but continues to rely on the conventional structure-from-motion for initial scene setup.

## 3 Method

As a classic task in computer vision, taking a sequence of plain 2D RGB images $\{I_i\}_{i=1}^N, I_i \in \mathbb{R}^{H \times W \times 3}$ as input, our goal is to reconstruct the 3D geometry using 3D Gaussian splatting and learn the extrinsic and intrinsic parameters of the cameras for the corresponding images simultaneously.

### 3.1 Preliminary: 3D Gaussian Splatting

Kerbl et al. (2023) proposed a novel representation for 3D geometry that can be rendered with photorealistic quality and high performance by leveraging the conventional rasterization pipeline. The primitive is an ellipsoid formulated with a 3D Gaussian function:

$$G(\mathbf{x}) = \exp\left(-\tfrac{1}{2}\left(\mathbf{x} - \boldsymbol{\mu}\right)^T \Sigma^{-1} \left(\mathbf{x} - \boldsymbol{\mu}\right)\right). \tag{1}$$

Each 3D Gaussian is parameterized by a mean position $\boldsymbol{\mu}$ and a covariance matrix $\Sigma \in \mathbb{R}^{3 \times 3}$. The ellipsoid defined by Equation 1 is equivalent to a sphere that has been scaled along the coordinate axes and then rotated. Consequently, the covariance can be expressed as a composition of a scaling matrix $S \in \mathbb{R}^{3 \times 3}$ (typically diagonal) and a rotation $R \in \mathbb{R}^{3 \times 3}$, e.g. $\Sigma = RSS^T R^T$, which ensures that $\Sigma$ is positive semi-definite.

To render a 3D Gaussian splat (3DGS), it is projected onto the screen (canvas). Given the camera pose represented by a rigid-body transform $W \in \mathrm{SE}(3)$ and camera intrinsics $K \in \mathbb{R}^{3 \times 4}$, the mean

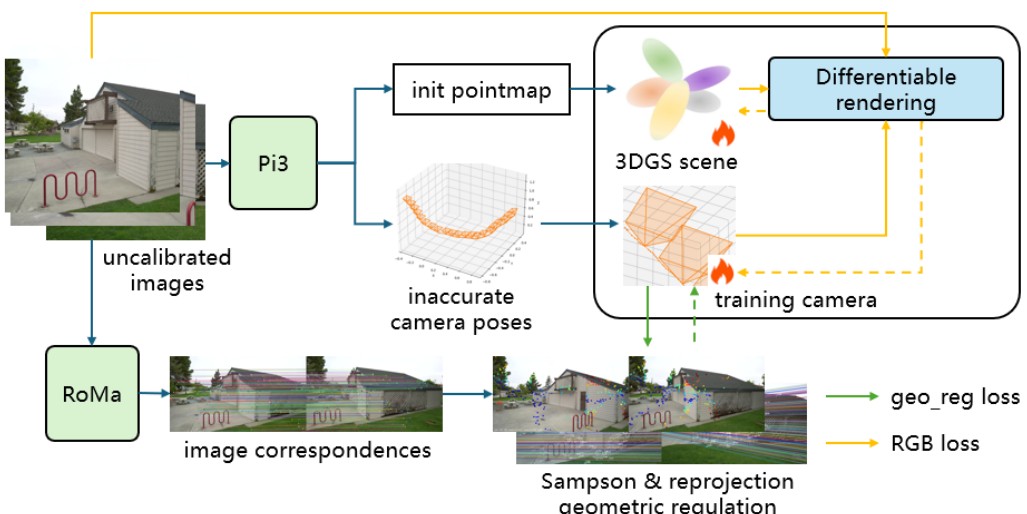

Figure 1: Overview of our Pi3GS pipeline. The geometry and the cameras are optimized simultaneously.

$\boldsymbol{\mu}$ is projected to image coordinates $\mathbf{x}_j^{\mathrm{2D}}$. The projected covariance is approximated by propagating the 3D covariance via $J$, the Jacobian of the camera projection matrix (Zwicker et al., 2001): $\Sigma^{\mathrm{2D}} = J\,W\,\Sigma\,W^T J^T$, where $J$ denotes the Jacobian of the projection at the Gaussian center. This approximation yields a 2D Gaussian in image space and enables rasterization without explicit 3D sampling, making the representation compatible with standard GPU rasterization pipelines.

The appearance of each Gaussian is represented using spherical harmonics: $\mathbf{c}_i = \mathrm{SH}_i(\mathbf{v})$, where $\mathbf{v} \in \mathbb{R}^3$ is the normalized view direction. Parameterized by a low-dimensional feature vector, the spherical harmonics basis compactly encodes low-frequency, view-dependent color variation.

The final pixel color is obtained by compositing the contributions of individual Gaussians in a front-to-back order. Denoting the opacity contribution of the $i$-th Gaussian by $\alpha_i = o_i\,G_i(x)$ (where $o_i$ accounts for per-splat opacity), the color is

$$\mathbf{C} = \sum_{i=1}^{N} \mathbf{c}_i\,\alpha_i \prod_{j=1}^{i-1} \left(1 - \alpha_j\right), \tag{2}$$

which corresponds to standard alpha compositing applied to the 2D Gaussian splats.

## 3.2 PI3DGS PIPELINE

Our objective is to reconstruct a high-fidelity 3D geometry and shading (3DGS) representation from a collection of uncalibrated multi-view RGB images. Conventional 3DGS reconstruction typically requires accurate camera intrinsics and poses, which restricts its applicability for many downstream tasks. Pi3 (Wang et al., 2025c) is an image-matching model pre-trained on extensive 3D datasets and thus embodies a strong 3D prior. Given a sequence of 2D images, Pi3 predicts an initial 3D point cloud and estimates camera intrinsics via reprojection error minimization. Leveraging a large neural model, Pi3 can produce satisfactory initial point clouds and camera poses in many cases. However, in the absence of explicit regularization on camera poses, these initializations may contain non-negligible noise, particularly for outdoor scenes with large depth ranges.

Because Pi3 does not provide dense pixel correspondences, its initialization alone is insufficient to establish a robust geometric constraint. We therefore employ RoMa (Edstedt et al., 2024), an efficient and robust image-matching model, to extract pairwise correspondences across the image sequence and thereby strengthen the geometric supervision.

**Camera modeling.** We represent each camera pose subject to refinement as the composition of an initial pose and a trainable corrective transformation:

$$T_i = \Delta T_i \, \hat{T}_i, \tag{3}$$

where $\hat{T}_i$ denotes the Pi3 initialization and $\Delta T_i$ is a learnable refinement. To reduce coupling between rotation and translation during optimization, we parameterize $\Delta T_i$ by decoupling its rotation and translation components: the rotation is represented in the Lie algebra $\mathfrak{so}(3)$, while the translation is represented in $\mathbb{R}^3$ and multiplied by a learnable scalar $s$. The scalar $s$ enhances the robustness of translation optimization across scenes with varying absolute scales.

**Initialization.** For each input image $I_i \in \{I_i\}_{i=1}^N$, Pi3 produces a local point map $\mathbf{X}_i^N \in \mathbb{R}^{H \times W \times 3}$ and an approximate camera pose $\hat{T}_i = [R_i \mid \mathbf{t}_i]$. The intrinsic matrix $K_i$ is not provided and must be estimated. Given the approximate extrinsics $\hat{T}_i$, we estimate $K_i$ by minimizing the reprojection error between the 3D points and their observed image projections.

Let $\mathbf{x}_j^{\text{3D}} \in \mathbf{X}_i^N$ be a 3D point and $\mathbf{x}_j^{\text{2D}} \in \mathbb{R}^2$ its observed image location. In homogeneous coordinates the projection relation is

$$\bar{\mathbf{x}}_j^{\text{2D}} \sim K_i \, [R_i \mid \mathbf{t}_i] \, \bar{\mathbf{x}}_j^{\text{3D}}, \tag{4}$$

where $\bar{\mathbf{x}}_j^{\text{2D}}$ and $\bar{\mathbf{x}}_j^{\text{3D}}$ denote homogeneous coordinates, $K_i$ is the intrinsic matrix, and $[R_i \mid \mathbf{t}_i]$ are the extrinsics from Pi3. The intrinsics are recovered by minimizing the reprojection error:

$$K_i = \arg\min_K \sum_j \left\| \pi\left( K \, [R_i \mid \mathbf{t}_i] \, \bar{\mathbf{x}}_j^{\text{3D}} \right) - \mathbf{x}_j^{\text{2D}} \right\|^2, \tag{5}$$

where $\pi(\cdot)$ denotes perspective division. This problem can be linearized and solved in a least-squares sense to obtain an initial estimate of $K_i$. If all cameras are assumed to share identical intrinsics, the shared intrinsic matrix $K$ is estimated jointly over the entire sequence.

### 3.3 ALIGNMENT WITH SMOOTHED GRADIENT

3D Gaussian blurring can be formulated as a smooth operator acting on the scale component of a 3DGS. Concretely,

$$G_i(\mathbf{x}) = \sqrt{\frac{|\Sigma|}{|\Sigma + \epsilon_i^2 I|}} \exp\left( -\tfrac{1}{2}(\mathbf{x} - \boldsymbol{\mu})^\top (\Sigma + \epsilon_i^2 I)^{-1} (\mathbf{x} - \boldsymbol{\mu}) \right). \tag{6}$$

This operation increases the effective covariance of each Gaussian and attenuates its peak amplitude, thereby simulating a spatial spread of the underlying density.

To ensure consistent blurring of projected Gaussians in image space, we set the blur magnitude $\epsilon_i$ proportionally as

$$\epsilon_i = \epsilon_{\text{base}} \cdot d_i, \tag{7}$$

where $d_i$ denotes the depth of the Gaussian center in camera coordinates (i.e., its $z$ coordinate). This depth-proportional scaling balances sampling density across the image plane and applies a scale-appropriate blur to each Gaussian, yielding more stable and consistent training dynamics.

For 2D supervision, rendered images are smoothed by convolution with a 2D Gaussian kernel. By scheduling the intensity of both the 3D and 2D blurring terms, we realize a coarse-to-fine alignment strategy: stronger blurring is applied early to suppress high-frequency noise, and the blur is gradually reduced to handle finer signals. Following the analysis of TensorRF (Chen et al., 2024), this controlled blurring of high-frequency content promotes robust and stable camera-pose alignment and regularizes the frequency spectrum of the 3DGS primitives.

### 3.4 IMAGE MATCHING BASED GEOMETRIC REGULATION

Although Pi3 has provided a strong prior on various 3D information including point cloud, camera poses. However, according to our observation, several problems arise in supplying these initializations directly to the 3DGS reconstruction pipeline. Pi3 does not output correspondence like other

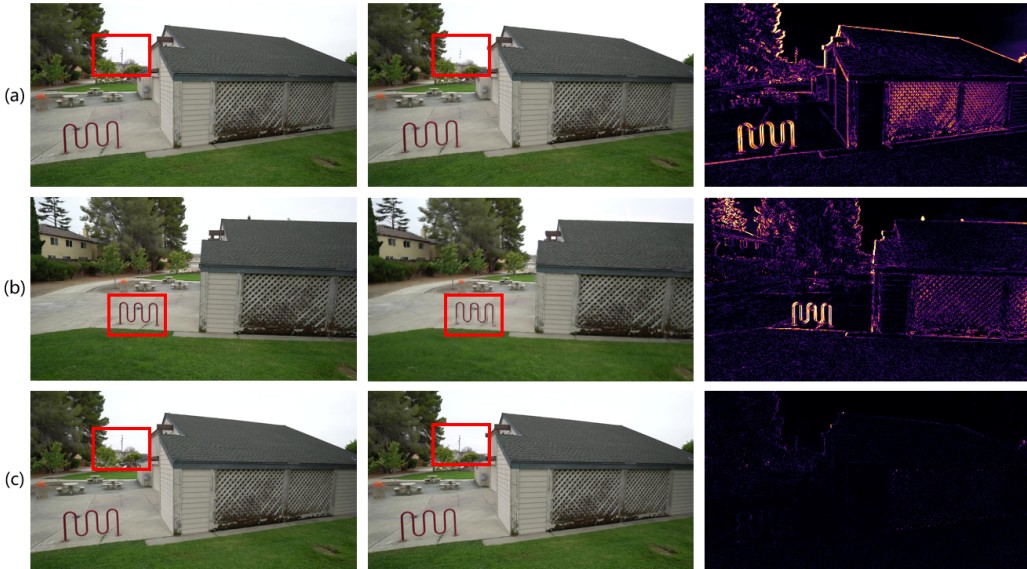

Figure 2: Effect of camera perturbations in the absence of geometric regularization. From left to right: ground truth, rendered result, and error map. From top to bottom: (a) an outlier camera that fails to align the ground-truth and rendered views; (b) artifacts resulting from misaligned cameras; (c) results produced when our geometric constraint is applied. Zoom in for finer details.

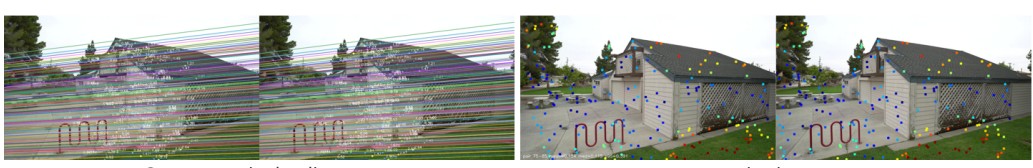

Sampson epipolar distance                                    reprojection error

Figure 3: Visualization of epipolar and reprojection geometry. When camera motion exhibits a small baseline, the epipolar lines become nearly parallel, which reduces the effectiveness of the epipolar constraint (left). Therefore, we report the reprojection error as a more direct measure (right). Colors map error magnitude from blue (low) to red (high).

large 3D models, we exploit RoMa (Edstedt et al., 2024), a robust but accurate pre-trained image matching model to acquire pairwise correspondences.

Based on the correspondences, we implement our framework with both the Sampson epipolar distance and reprojection error to maintain the relationship between the cameras. This improves the global stability for the cameras in searching for blurred phrases. Specifically, the former is a robust and symmetric version of the epipolar distance. The latter is a more sensitive regulation based on image matching, which is more effective for low-baseline scenarios.

$$\mathcal{L}_{\text{match}} = \frac{\lambda_{\text{epi}}}{\sum_{i=1}^{N} c_i} \sum_{i=1}^{N} c_i \cdot d_{\text{epi}}\left(\mathbf{x}_i, \mathbf{x}_i'\right) + \frac{\lambda_{\text{reproj}}}{\sum_{i=1}^{N} c_i} \sum_{i=1}^{N} c_i \cdot d_{\text{reproj}}\left(\mathbf{x}_i, \mathbf{x}_i'\right) \tag{8}$$

Here, $d_{\text{epi}}\left(\mathbf{x}_i, \mathbf{x}_i'\right)$ and $d_{\text{reproj}}\left(\mathbf{x}_i, \mathbf{x}_i'\right)$ represent the symmetric Sampson distance and reprojection error of correspondence $\left(\mathbf{x}_i, \mathbf{x}_i'\right)$ in pixel coordinates.

Additionally, regularly resetting the opacity is a common scheme to reduce artifacts by removing wrongly generated floater Gaussians. This operation produces a large fluctuation to the RGB gradient, causing unsteadiness or even mis-convergence to the cameras.

## 3.5 TOTAL TRAINING LOSS

The overall training objective comprises three terms: the 3DGS rendering loss, a geometric regularization term based on image matching, and a scale regularization term:

$$\mathcal{L}_{\text{total}} = \mathcal{L}_{\text{rgb}} + \mathcal{L}_{\text{match}} + \mathcal{L}_{\text{scale\_reg}}. \tag{9}$$

The photometric rendering loss follows standard practice and combines an L1 term with a structural similarity term:

$$\mathcal{L}_{\text{rgb}} = (1 - \lambda_{\text{SSIM}}) \, \mathcal{L}_1 + \lambda_{\text{SSIM}} \, \mathcal{L}_{\text{SSIM}}, \tag{10}$$

where $\lambda_{\text{SSIM}} = 0.2$ balances the two components. To prevent degenerate or ill-conditioned Gaussian primitives, we introduce a scale regularization term that constrains the aspect ratio of each Gaussian's principal axes:

$$\mathcal{L}_{\text{scale\_reg}} = \frac{1}{N} \sum_{i=1}^{N} \left[ \max\left( \frac{\max(S_i)}{\min(S_i)}, r \right) - r \right]_+, \tag{11}$$

where $S_i \in \mathbb{R}^d$ is the vector of scale parameters (the lengths of the principal axes) for the $i$-th Gaussian, $r > 1$ denotes the allowable maximum aspect ratio, and $[\cdot]_+ = \max(\cdot, 0)$ ensures the term is nonnegative.

Table 1: Novel view evaluation on Tanks and Temples dataset. The best results are highlighted in bold.

| Method | Ours | | | KeyGS | | | CF-3DGS | | | Nope-NeRF | | |
|---|---|---|---|---|---|---|---|---|---|---|---|---|
| | PSNR↑ | SSIM↑ | LPIPS↓ | PSNR | SSIM | LPIPS | PSNR | SSIM | LPIPS | PSNR | SSIM | LPIPS |
| Chruch | **31.61** | **0.95** | **0.04** | 30.62 | 0.92 | 0.06 | 30.23 | 0.93 | 0.11 | 25.17 | 0.73 | 0.39 |
| Barn | 33.61 | 0.94 | **0.03** | **34.25** | **0.95** | 0.04 | 31.23 | 0.90 | 0.10 | 26.35 | 0.69 | 0.44 |
| Museum | **35.85** | **0.97** | **0.01** | 33.46 | 0.94 | 0.03 | 29.91 | 0.91 | 0.11 | 26.77 | 0.76 | 0.35 |
| Family | **34.54** | **0.97** | **0.02** | 33.05 | 0.95 | 0.04 | 31.27 | 0.94 | 0.07 | 26.01 | 0.74 | 0.41 |
| Horse | **35.02** | **0.97** | **0.02** | 33.65 | 0.96 | 0.03 | 33.94 | 0.96 | 0.05 | 27.64 | 0.84 | 0.26 |
| Ballroom | **36.63** | **0.98** | **0.01** | 33.70 | 0.95 | 0.02 | 32.47 | 0.96 | 0.07 | 25.33 | 0.72 | 0.38 |
| Francis | **35.07** | **0.96** | **0.05** | 34.45 | 0.93 | 0.08 | 32.72 | 0.91 | 0.14 | 29.48 | 0.80 | 0.38 |
| Ignatius | **33.29** | **0.96** | **0.03** | 30.85 | 0.92 | 0.06 | 28.43 | 0.90 | 0.09 | 23.96 | 0.61 | 0.47 |
| Mean | **34.45** | **0.96** | **0.03** | 33.00 | 0.94 | 0.05 | 31.28 | 0.93 | 0.09 | 26.34 | 0.74 | 0.39 |

## 4 EXPERIMENTS

### 4.1 EXPERIMENTAL SETUP

**Datasets and Metrics.** We evaluate our method on two datasets. The Tanks and Temples dataset (Knapitsch et al., 2017) contains large-scale reconstruction scenarios covering both indoor and outdoor environments. CO3Dv2 (Reizenstein et al., 2021) comprises numerous multi-view captures of common objects. We compare against CF-3DGS (Fu et al., 2024), NoPe-NeRF (Bian et al., 2023), and keyGS (Chang et al., 2025).

Evaluation addresses both novel-view synthesis and camera-pose estimation; an ablation study quantifies the contribution of individual components. For novel view synthesis we report PSNR, SSIM, and LPIPS as measures of photometric and perceptual quality. Camera pose accuracy is assessed using translational and rotational relative pose error ($\text{RPE}_t$ and $\text{RPE}_r$) as well as the absolute trajectory error (ATE).

**Implementation Details.** Our implementation is built on Nerfstudio (Tancik et al., 2023) with the gsplat extension (Ye et al., 2025), which provides a flexible framework for constructing a custom 3DGS reconstruction pipeline. To jointly optimize the 3DGS representation and camera poses, we

tune several hyperparameters. For camera optimization we employ exponential decay schedules, using base learning rates of $1 \times 10^{-5}$ and $5 \times 10^{-5}$ for rotation and translation, respectively, to balance their relative update magnitudes.

## 4.2 RESULTS AND COMPARISONS

**Novel view synthesis.** We first evaluate on a Tanks and Temples version preprocessed by CF-3DGS (Fu et al., 2024), which provides densely sampled frames extracted from short video clips. As reported in Table 1, our approach achieves superior visual-quality metrics in the majority of cases compared to the baselines.

On the CO3Dv2 dataset, we further evaluate the effectiveness of our approach under large camera motions. This dataset provides frames that capture objects over camera trajectories exceeding 180 degrees in most cases. As reported in Table 2, our method demonstrates robust performance and maintains competitive visual quality for novel view synthesis under these challenging conditions.

Table 2: Novel view evaluation on CO3Dv2 dataset. The best results are highlighted in bold.

| Method | Ours | | | KeyGS | | | CF-3DGS | | |
|---|---|---|---|---|---|---|---|---|---|
| | PSNR↑ | SSIM↑ | LPIPS↓ | PSNR | SSIM | LPIPS | PSNR | SSIM | LPIPS |
| Apple | **33.98** | **0.94** | **0.07** | 33.53 | **0.94** | **0.07** | 29.69 | 0.89 | 0.29 |
| Bench | **31.75** | **0.92** | **0.10** | 26.35 | 0.73 | 0.30 | 26.21 | 0.73 | 0.32 |
| Hydrant | **28.66** | **0.91** | **0.07** | 25.33 | 0.80 | 0.15 | 22.14 | 0.64 | 0.34 |
| SkateBoard | **34.51** | **0.94** | **0.11** | 32.74 | 0.93 | 0.16 | 27.24 | 0.85 | 0.30 |
| Teddybear | **33.89** | **0.94** | **0.09** | 32.67 | 0.93 | **0.09** | 27.75 | 0.86 | 0.20 |
| Mean | **32.56** | **0.93** | **0.09** | 30.12 | 0.87 | 0.15 | 26.61 | 0.79 | 0.29 |

**Camera pose estimation.** We include KeyGS for reference only, since it is initialized with COLMAP results, which serves as ground-truth camera poses in our evaluation. As reported in Table 3, our method achieves the best performance in estimating the translation component of camera poses, which explains the superior visual quality observed in the novel view synthesis.

Table 3: Camera poses estimation on Tanks and Temples dataset. The best results are highlighted in bold except "KeyGS*" method as it is initialize with COLMAP sequence mode.

| Method | ours | | | Nope-NeRF | | | CF-3DGS | | | KeyGS* | | |
|---|---|---|---|---|---|---|---|---|---|---|---|---|
| | $RPE_t$↓ | $RPE_r$↓ | ATE↓ | $RPE_t$ | $RPE_r$ | ATE | $RPE_t$ | $RPE_r$ | ATE | $RPE_t$ | $RPE_r$ | ATE |
| Church | **0.007** | 0.034 | **0.000** | 0.008 | 0.018 | 0.002 | 0.034 | **0.008** | 0.008 | 0.006 | 0.013 | 0.000 |
| Barn | **0.007** | 0.036 | **0.000** | 0.034 | 0.034 | 0.003 | 0.046 | **0.032** | 0.004 | 0.008 | 0.016 | 0.001 |
| Museum | **0.024** | **0.039** | 0.001 | 0.052 | 0.215 | 0.005 | 0.207 | 0.202 | 0.020 | 0.025 | 0.025 | 0.002 |
| Family | **0.012** | 0.033 | **0.000** | 0.022 | 0.024 | 0.002 | 0.047 | **0.015** | 0.001 | 0.012 | 0.012 | 0.000 |
| Horse | **0.082** | 0.037 | **0.001** | 0.112 | 0.057 | 0.003 | 0.179 | **0.017** | 0.003 | 0.078 | 0.002 | 0.001 |
| Ballroom | **0.015** | 0.033 | **0.000** | 0.037 | 0.024 | 0.003 | 0.041 | **0.018** | 0.002 | 0.015 | 0.014 | 0.000 |
| Francis | **0.008** | 0.036 | **0.001** | 0.029 | 0.154 | 0.006 | 0.057 | **0.009** | 0.005 | 0.007 | 0.016 | 0.001 |
| Ignatius | **0.008** | 0.033 | **0.001** | 0.033 | 0.032 | 0.005 | 0.026 | **0.005** | 0.002 | 0.001 | 0.010 | 0.001 |
| **mean** | **0.020** | **0.035** | **0.000** | 0.0409 | 0.0698 | 0.0036 | 0.080 | 0.038 | 0.006 | 0.019 | 0.014 | 0.001 |

## 4.3 ABLATION STUDY

In this section, we highlight the contributions of the key components and strategies of our proposed approach. All experiments were conducted on the Tanks and Temples dataset. As summarized in Table 4, gradient smoothing (GS) and geometry regularization (Geo-Reg) are evaluated both

Figure 4: Comparison of image-matching approaches for correspondence extraction. For scenes with larger inter-view camera motion, we employ RoMa (Edstedt et al., 2024) to obtain more reliable correspondences.

individually and jointly. The results show that each component yields measurable improvements in visual quality on its own and that applying both components together produces further gains across the evaluated quantitative metrics.

Table 4: Ablation study of model components.

| GS | Geo-Reg | PSNR↑ | SSIM↑ | LPIPS↓ |
|----|---------|-------|-------|--------|
|    |         | 33.98 | 0.93  | 0.08   |
| ✓  |         | 34.12 | 0.94  | 0.05   |
|    | ✓       | 34.24 | 0.93  | 0.07   |
| ✓  | ✓       | 34.45 | 0.96  | 0.03   |

According to our analysis, initialization plays a key role in reconstruction quality. The 3DGS representation is sensitive to initial camera estimates and has limited ability to correct erroneously initialized Gaussian ellipsoids. Table 5 compares several initialization strategies: COLMAP, representing a conventional SfM pipeline, and MASt3r, representing a learning-based dense reconstruction method. The results demonstrate that our proposed initialization outperforms these baselines across the evaluated metrics in terms of both reconstruction accuracy and visual fidelity.

Table 5: Ablation study of initialization strategies. The "MASt3r" initialization employs correspondences extracted by MASt3r.

| Method | Geo-Reg | PSNR↑ | SSIM↑ | LPIPS↓ |
|--------|---------|-------|-------|--------|
| COLMAP |         | 31.76 | 0.91  | 0.09   |
| MASt3r | ✓       | 32.73 | 0.93  | 0.05   |
| Ours   |         | 34.12 | 0.94  | 0.05   |
| Ours   | ✓       | 34.45 | 0.96  | 0.03   |

## 5 CONCLUSION

3D reconstruction is a fundamental problem in computer vision. We propose Pi3DGS, an efficient yet robust framework that jointly optimizes camera poses and the 3D Gaussian Splatting (3DGS) representation from uncalibrated multi-view images. We investigate the impact of camera pose optimization on 3DGS reconstruction and further introduce two key contributions: (i) leveraging the large-scale 3D model Pi3 to enhance representation quality, and (ii) a gradient-smoothed pipeline with regularization guided by image matching. Experimental results demonstrate that our framework achieves superior performance in both novel view synthesis and pose estimation compared to existing approaches.

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

# A  APPENDIX

## A.1  USE OF LLMS

With respect to the involvement of large language models (Liu et al., 2024; Achiam et al., 2023) in this research, their use was limited to grammatical review and stylistic polishing to increase formality. They were not employed to generate scientific content, design experiments, analyze data, or interpret results; all substantive intellectual contributions were made by the authors.

