# OpenReview forum: "Pi3DGS: Robust Joint Optimization of Camera Poses and 3DGS from Uncalibrated Images"
_ICLR.cc/2026/Conference — ICLR 2026 Conference Withdrawn Submission_

### Official Review · Reviewer_8JWF · 2025-10-21

**Soundness:** 4
**Presentation:** 4
**Contribution:** 3
**Rating:** 8
**Confidence:** 5

**Summary:**

This work presents a robust, SfM-free neural rendering framework for 3D reconstruction from uncalibrated RGB images. The proposed method jointly optimizes scene geometry and camera poses by leveraging pre-trained 3D feed-forward models within a coarse-to-fine alignment pipeline. Using Pi3 for initialization and incorporating multi-level geometric regularization, the approach enhances pose stability and reconstruction quality, achieving competitive performance in both novel view synthesis and pose estimation across diverse datasets.

**Strengths:**

[S1] **Clear motivation:** The paper presents a well-defined and convincing motivation that clearly identifies the gap in existing approaches. Their argument for addressing the reliance on traditional SfM pipelines is particularly persuasive and timely. The proposed solution is conceptually simple yet logically derived from their stated motivation, making the paper easy to follow.

[S2] **Astonishing performance**: The method achieves remarkable performance improvements, surpassing prior state-of-the-art methods such as KeyGS. Although it leverages recent geometric foundation models, the performance gain is substantial and well-justified through experiments. Its ability to outperform even COLMAP-based initialization highlights the strong potential of feed-forward architectures over conventional SfM pipelines.

[S3] **Readily readable writing**. The manuscript is exceptionally well-written, presenting technical details with clarity and precision. Each section flows logically, allowing readers to easily follow the methodology and experimental reasoning. Figures, equations, and explanations are consistently well-aligned, contributing to a highly readable presentation.

[S4] **Well organized ablation studies**. The ablation studies are comprehensive and systematically designed to validate the contribution of each component. They effectively demonstrate how different architectural or training choices influence overall performance. The inclusion of detailed quantitative and qualitative analyses enhances the paper’s scientific rigor and credibility.

**Weaknesses:**

I believe this paper is well-written, and I have no further questions regarding the experiments, as the authors have provided sufficient details and clarity throughout.

[W1] In the introduction, the authors mention that feed-forward models struggle with scenes exhibiting large depth ranges. However, the paper does not provide concrete demonstrations of how such depth variations are handled by their model. It would strengthen the paper if the authors included experiments evaluating depth estimation performance of their optimized model and compared the depth predictions before and after optimization (e.g., between Pi3 initialization and the final results).

[W2] Figure 3 is somewhat difficult to interpret, even for readers familiar with reprojection error and epipolar constraints. I suggest adding visual aids such as arrows to explicitly illustrate the direction and magnitude of reprojection errors, which would improve the clarity of the figure.

[W3] I am also curious about the contribution of each term in the matching loss. The authors state that reprojection and epipolar errors play different roles, but this claim is not empirically supported. Including an ablation study or additional analysis to demonstrate the effect of each term would make the argument more convincing

Expressions
- (L34-L36) NeRF accepts 3D sample points and view directions as input and predicts radiance and density via a neural network, which are then integrated by volume rendering.
    - NeRF does not accept 3D points and view directions, but the network of NeRF does. It would be clear if the authors clarify NeRF’s network accepts 3D sample points and view directions to predict colors, and its volumetric rendering conducts alpha-blending over the predicted colors.
- (L39-L40) require accurate camera intrinsics and extrinsics for the input images. These parameters …
    - No mention about the parameters in the previous sentence. Recommend to changes as ‘require accurate camera intrinsic and extrinsic **parameters**  for the input image”
- (L43-44) across diverse conditions.
    - This is slightly misleading since VGGT and MASt3R strongly assume that images are captured in identical condition.
- (L69-73) In contrast, … local groupings.
    - What does ‘this pipeline limits the number of views that can be jointly optimized per step?’ mean? Is it because of the limited VRAM?
    - I think this part is crucial for overall flow of the introduction. I suggest to add more descriptions in this part.

**Questions:**

No specific questions for the paper. Overall questions and suggestions are in the Weakness section.

---

### Official Review · Reviewer_Fczn · 2025-10-31

**Soundness:** 3
**Presentation:** 3
**Contribution:** 3
**Rating:** 6
**Confidence:** 4

**Summary:**

This work does per-scene 3DGS optimization without known camera parameters based on the recent geometry foundation model Pi3. The Gaussians are initialized by the predicted point map from Pi3. The low pass Gaussian filter is designed to be Gaussian dependent based on their initial estimated depth from Pi3. Additional feature matching model is employed to derive geometric regularization for camera pose optimization. Results on real-world and synthetic objects datasets both show significant improvement.

**Strengths:**

The quality improvement is very good. The gradient smoothing based on prior depth is very useful and the coarse-to-fine training strategy with it are intuitively helpful. Its improvement on LPIPS is also significant. I think it may become a standard strategy for future geometric prior dependent 3DGS training.

**Weaknesses:**

A more straightforward alternative for gradient smoothing is simply initialize Gaussian scale parameters proportional to the prior depth. As such, the model may can learn itself how to adjust its level-of-details instead of a heuristic schedule. I wonder the benefit of the proposed gradient smoothing comparing to this naive version.

Is the pruning and densification strategy change? The current initialization have a much denser point cloud due to the point map prediction from Pi3. When we have a denser input views, many of the initial points may be redundant can cause training slow. How does the proposed system deal with this?

It seems that the improvement over previous work is coupled with the stronger Pi3 backbone. If the same backbone are used following previous work, what will be the quantitive improvement then?

**Questions:**

What is the training time, FPS, and memory cost? As the proposed method starts from a dense point cloud. I think the cost may linearly increase with number of input images?

---

### Official Review · Reviewer_cdWX · 2025-11-01

**Soundness:** 2
**Presentation:** 3
**Contribution:** 2
**Rating:** 4
**Confidence:** 4

**Summary:**

This paper presents Pi3DGS, an SfM-free method for jointly optimizing 3D Gaussian Splatting (3DGS) representations and camera poses directly from uncalibrated images. The method utilizes the Pi3 model for initialization. To stabilize the optimization, it introduces a gradient-smoothing strategy to manage the 3DGS geometry and a separate geometric regularization based on image matching to globally constrain and refine camera poses. The authors report competitive performance on both novel view synthesis and camera pose estimation benchmarks.

**Strengths:**

- The paper is clearly written and easy to follow.

- The framework demonstrates competitive or state-of-the-art performance, outperforming other SfM-free methods in both novel view synthesis and, critically, camera pose estimation on standard benchmarks.

**Weaknesses:**

- The method's success is contingent upon two separate, large-scale pre-trained models: Pi3 for initialization and RoMa for geometric regularization. This creates a strong dependency, and the paper does not sufficiently explore the method's robustness or failure modes when these upstream models provide poor priors.

- The novelty of this work is primarily incremental. While the specific combination of pipeline components is novel, the core stabilization techniques are merely adaptations of existing ideas. Both coarse-to-fine gradient smoothing and image-matching-based geometric constraints have been previously explored for joint pose optimization

**Questions:**

- The proposed pipeline introduces the overhead of running two separate, large-scale neural models (Pi3 and RoMa) before the main 3DGS optimization. How does the total, end-to-end computational time of this method compare to a traditional SfM (e.g., COLMAP) + 3DGS pipeline? Is the gain in robustness worth a potentially significant increase in runtime?

- The paper rightly emphasizes that 3DGS is sensitive to initial camera estimates. While Pi3 provides a strong start, how robust is the optimization framework to the quality of this initialization?

- Given that some recent pose-free methods are moving towards fully end-to-end joint optimization [1], this paper's approach still relies on two separate foundation models (Pi3 and RoMa) for initialization and supervision. What are the specific advantages of this decoupled, multi-stage approach compared to a truly end-to-end framework?

[1] GGRt: Towards Pose-free Generalizable 3D Gaussian Splatting in Real-time. ECCV 2024.

---

### Official Review · Reviewer_AsCk · 2025-11-01

**Soundness:** 2
**Presentation:** 1
**Contribution:** 2
**Rating:** 2
**Confidence:** 4

**Summary:**

This paper presents Pi3DGS, a framework for reconstructing 3D Gaussian Splatting scenes directly from uncalibrated RGB images, removing the need for SfM preprocessing. The proposed system leverages Pi3 for initialization and then jointly optimizes camera poses and the 3DGS representation. The key contributions are:

1. A gradient-smoothing strategy that stabilizes early training and mitigates the effects of inaccurate initialization.
2. An image-matching-based geometric regularization that enforces multi-view consistency and improves pose refinement.

Experiments on the Tanks and Temples and CO3Dv2 datasets show improved performance in novel view synthesis and camera pose estimation over existing methods such as CF-3DGS, KeyGS, and Nope-NeRF. The paper also includes ablation studies that demonstrate the impact of gradient smoothing and geometric regularization components.

**Strengths:**

1. Well-motivated and practically relevant problem. The paper addresses the challenging and important scenario of SfM-free 3D reconstruction, where camera calibration is unavailable or unreliable.
2. Comprehensive quantitative evaluation with consistent improvements across multiple benchmarks and metrics, demonstrating the method’s practical utility and robustness.

**Weaknesses:**

1. Limited Novelty / Primarily Engineering Integration. The proposed method mainly combines existing techniques rather than introducing new algorithmic insights:
    - Pi3 for initialization.
    - Gradient smoothing inspired by Mip-Splatting and TensorRF.
    - Image-matching regularization following ideas from PoRF.

    Therefore, the contribution is largely engineering-oriented rather than conceptually novel.

2. No Evaluation of Pi3 Initialization Quality. The framework begins from Pi3 initialization and further optimizes camera poses, but the paper does not evaluate or quantify the accuracy of Pi3’s initial poses. As a result, it is unclear how much improvement the proposed joint optimization actually achieves over the baseline Pi3 output.
3. Weak Presentation and Incomplete Experimental Reporting. The paper’s presentation quality is incomplete:
    - Figures are poorly referenced and insufficiently explained in the text, reducing interpretability.
    - There is a lack of qualitative visual comparisons and analysis of rendered results, which limits understanding of perceptual improvements.
    - The discussion of visual examples and error maps is minimal and lacks detailed interpretation.
4. Lack of Comparison with InstantSplat. The paper does not include a comparison with InstantSplat, a closely related method. Although InstantSplat is limited by DUSt3R and primarily designed for few-shot image scenarios, a comparative evaluation under such few-shot settings would still be valuable. Including this comparison would clarify the relative strengths of Pi3DGS, especially in low-view regimes where robustness to limited input data is critical.

**Questions:**

1. How sensitive is the performance to the quality of Pi3’s initial camera poses?
2. Are the 2D and 3D Gaussian blurring schedules adaptive or fixed across datasets?
3. Could the image-matching constraint overfit in repetitive or textureless regions?

---

### Note · Authors · 2025-11-12

I have read and agree with the venue's withdrawal policy on behalf of myself and my co-authors.